# Isavuconazole for COVID-19-Associated Invasive Mold Infections

**DOI:** 10.3390/jof8070674

**Published:** 2022-06-28

**Authors:** Nadir Ullah, Chiara Sepulcri, Malgorzata Mikulska

**Affiliations:** 1Department of Health Sciences (DISSAL), University of Genova, 16132 Genova, Italy; nadir.ullah@uvas.edu.pk (N.U.); chiara.sepulcri@gmail.com (C.S.); 2Division of Infectious Diseases, Ospedale Policlinico San Martino, 16132 Genova, Italy

**Keywords:** isavuconazole, CAPA, CAM, invasive fungal infection, COVID-19, aspergillosis, mucormycosis, RRT, ECMO, TDM

## Abstract

Isavuconazole is a broad-spectrum antifungal drug recently approved as a first-line treatment for invasive aspergillosis and as a first or alternative treatment for mucormycosis. The purpose of this review was to report and discuss the use of isavuconazole for the treatment of COVID-19-associated aspergillosis (CAPA), and COVID-19-associated mucormycosis (CAM). Among all studies which reported treatment of CAPA, approximately 10% of patients were reportedly treated with isavuconazole. Considering 14 identified studies that reported the use of isavuconazole for CAPA, isavuconazole was used in 40% of patients (95 of 235 treated patients), being first-line monotherapy in over half of them. We identified six studies that reported isavuconazole use in CAM, either alone or in combination therapy. Overall, isavuconazole was used as therapy in 13% of treated CAM patients, frequently as combination or sequential therapy. The use of isavuconazole in CAPA and CAM is complicated by the challenge of achieving adequate exposure in COVID-19 patients who are frequently obese and hospitalized in the ICU with concomitant renal replacement therapy (RRT) or extracorporeal membrane oxygenation (ECMO). The presence of data on high efficacy in the treatment of aspergillosis, lower potential for drug–drug interactions (DDIs) and for subtherapeutic levels, and no risk of QT prolongation compared to other mold-active azoles, better safety profile than voriconazole, and the possibility of using an intravenous formulation in the case of renal failure are the advantages of using isavuconazole in this setting.

## 1. Introduction

SARS-CoV-2 virus, the causative agent of COVID-19, first appeared in late 2019 and has since spread around the world. Patients with COVID-19 can develop severe pneumonia, requiring hospitalization and intensive care unit (ICU) admission and ventilatory support. Invasive fungal infections have been reported in ICU-admitted patients, particularly with respiratory failure due to influenza; with up to 14% developing invasive aspergillosis in this setting [1]. COVID-19 was also found to predispose ICU patients to invasive fungal infections, both due to insults in the lungs resulting in the release of danger-associated molecular pattern (DAMPs) and the impaired local immune response and dysfunctional mucociliary activity, and due to immunomodulating treatment, mainly with high dose steroid therapy, used to counteract COVID-19-related inflammation [1,2,3,4,5].

The most frequently reported invasive fungal infections in patients with COVID-19 are COVID-19 associated pulmonary aspergillosis (CAPA), and COVID-19-associated mucormycosis (CAM). 

The incidence of CAPA has been estimated as ranging from 2 to 33% in ICU cohort studies, but the incidence may vary among the centers [6]. CAPA has proven difficult to diagnose, and was reported to have a significant mortality rate, with all-cause mortality ranging from 33% to 80%, although precise CAPA-attributable mortality is very difficult to establish [7]. The European Confederation of Medical Mycology (ECMM) and the International Society for Human and Animal Mycology (ISHAM) collaborated to create diagnostic criteria and care protocols for patients with CAPA and COVID-19-associated acute respiratory distress syndrome (ARDS) [8]. These efforts should result in increased certainty of diagnosis and better opportunities to compare experiences from different centers. While CAPA has been a major focus among fungal infections during the COVID-19 pandemic [8], other mycoses have also been reported, such as candidiasis [2], and less common mold infections, such as fusariosis [9] and CAM [10,11,12].

Mucormycosis is an invasive fungal infection caused by a family of opportunistic molds known as mucoromycetes. Initially, CAPA received a high level of attention, as compared to other fungal infections such as CAM, until a large number of CAM cases was reported from India [13]. Indeed, the multicenter study in India reported a 2.1 times higher frequency of mucormycosis during the COVID-19 pandemic [14]. In 2021, the number of CAM has soared, and more than 45,000 cases of CAM and 4300 deaths have been reported in India [15]. The most probable factors responsible for CAM in India include the presence of uncontrolled diabetes, local epidemiology with high prevalence and high load of sporangiospores of Mucorales in the indoor and outdoor air, extensive and long-term usage of corticosteroids, and immunosuppression [15,16]. When comparing 233 cases of CAM from India and 42 from the rest of the world, diabetes and COVID-19 were the most frequent predisposing factors in both cohorts (respectively, 66% vs. 55% and 29% vs. 26%) while hematological malignancies and organ transplant recipients were less common in Indian cohorts (2% vs. 19%) [17]. The fatality rate of CAM cases reported globally was 62%, which was higher as compared to Indian cases (37%), possibly due to the predominance of rhino-orbital mucormycosis in India [17]. Pulmonary and disseminated mucormycosis have nonspecific clinical and radiological characteristics that may overlap with COVID-19 or with CAPA, resulting in a delayed diagnosis [10].

Various antifungal drugs available for first-line or salvage treatment of invasive mold infections include triazoles such as voriconazole, posaconazole, isavuconazole, itraconazole, amphotericin B, and echinocandins. Voriconazole and liposomal amphotericin B are probably the most frequently used in treatment of invasive mold infections in ICU patients, since they are approved as first-line therapy for, respectively, aspergillosis and mucormycosis [12,18]. Isavuconazole is a new mold-active triazole with a broader spectrum of activity compared to voriconazole, as in addition to Aspergilli, it is also active against Mucorales [19]. Isavuconazole showed similar efficacy and lower toxicity compared to voriconazole in a randomized trial in patients with invasive aspergillosis [20], and similar efficacy compared to liposomal amphotericin B in 21 patients with invasive mucormycosis [21]. Therefore, in 2015 it was approved for first-line treatment of aspergillosis and as a first or alternative treatment of mucormycosis by, respectively, the Food and Drug Administration (FDA) and European Medicines Agency (EMA); and subsequently also in other countries, including, for example, China in 2022. 

There are limited data on the real-life use of isavuconazole in patients with COVID-19-associated mold infections. The aim of this review was to report and discuss the use of isavuconazole for the treatment of COVID-19-related fungal infections, highlighting the frequency of use and the potential advantages and challenges.

## 2. Antifungal Activity of Isavuconazole

Isavuconazole has been shown to have good in vitro activity against *Aspergilli*, *Candida*, and most Mucorales. In 2017, antifungal activity of isavuconazole was determined against one of the largest recent collections of 958 Aspergillus isolates [22]. In 2021, Pfaller et al. reported data from 9 years of an antifungal surveillance program (2011–2019) during which 372 mold isolates from invasive infections were collected in Asia–Western Pacific and tested for susceptibility to isavuconazole, voriconazole, itraconazole, and posaconazole [23]. A total of 318 *Aspergillus* and 53 non-*Aspergillus* molds isolates were collected. Isavuconazole susceptibility testing against *Aspergillus* species has been performed since 2010. Isavuconazole, voriconazole, and posaconazole MIC_50_ and MIC_90_ against different *Aspergillus* species, together with EUCAST breakpoints, if available, are reported in Table 1. Of note, there has been no change in the MIC distribution over a 9-year period, although 5.7% of isolates were classified as non-wild type [23].

In vitro antifungal activity of isavuconazole was also determined against 72 clinical isolates of Mucorales and compared to other antifungal drugs including voriconazole, posaconazole, and amphotericin B using both EUCAST and CLSI methods. MIC_50_ (mg/L), of isavuconazole, voriconazole, and posaconazole, according to EUCAST susceptibility testing, against different Mucorales species were: 1 mg/L for *Lichtheimia corymbifera*, *L. ramose*, *Rhizopus oryzae*, and 0.5 mg/L for *R. microsporus* for isavuconazole; 16 mg/L for *Lichtheimia corymbifera*, and 8 mg/L for *L. ramose*, *Rhizopus oryzae*, *R. microsporus* for voriconazole; and 0.25 mg/L for all 4 species for posaconazole [24]. In fact, isavuconazole and posaconazole, had very good antifungal activity against all Mucorales species except for *Mucor circinelloides*, and overall 83–100% of isolates were potentially susceptible to isavuconazole (no breakpoints available), while 100% were potentially susceptible to amphotericin B [24]. 

Finally, Zheng et al. reported the in vitro activity of two novel triazoles: isavuconazole and ravuconazole, as well as 9 other antifungal drugs, against 84 clinical isolates of a wide range of dematiaceous fungus. Isavuconazole and ravuconazole have lower MIC values than voriconazole, posaconazole, and itraconazole. Another study reported excellent in vitro antifungal activity of isavuconazole against the *Madurella mycetomatis* causative agent of black grain (MICs ranging from ≤0.016 to 0.125 μg/mL) [25].

Azole-resistant *Aspergillus* isolates, mainly of *A. fumigatus*, are increasingly frequent worldwide, particularly in some geographical regions [26]. The resistance is thought to have been selected in the environment through the use of compounds similar to medical azoles in agricultural disease control, with mutations in the promoter region of the cyp51A gene (mainly TR_34_/L98H and TR_46_/Y121F/T289A) [27]. Less frequently, induction/selection of resistance to azoles might occur in vivo in patients receiving prolonged azole therapy, possibly at a suboptimal dosage. The presence of mutations of environmental origin usually confers the resistance to all available triazoles. In a multicenter study from an area with high prevalence of azole-resistance (the Netherlands), among 2266 patients with a positive *A. fumigatus* culture, 196 (8.6%) met the inclusion criteria of invasive aspergillosis [27]. Voriconazole resistance was observed in 37 (19%) patients, with the resistance frequency varying from 10% to 31% in different centers. Resistance to isavuconazole was also detected in all 14 patients with voriconazole-resistant strains in whom isavuconazole susceptibility was evaluated (MIC 8 or above in all of them); and in 30/37 cases (81%) *A. fumigatus* isolate was resistant to all triazoles. In 32 of 37 isolates (87%), mutations of environmental origin were detected (TR34/L98H in 18, TR46/Y121F/T289A in 14) [27]. 

In 2018, susceptibility to isavuconazole was specifically assessed with EUCAST broth microdilution methodology in 487 clinical isolates of *A. fumigatus* from a reference center in the Netherlands [28]. Among them, 279 isolates were phenotypically classified as wild-type based on epidemiological cut-offs of voriconazole, itraconazole, and posaconazole. With EUCAST breakpoint of 1 mg/L, resistance to isavuconazole was detected in 25/279 wild-type isolates, and 196/208 non-wild-type isolates (MIC of 4 or above in all isolates with cyp51A mutations of environmental origin). Monte Carlo simulation revealed that higher than standard doses of isavuconazole might achieve efficacy against isolates with MIC of 2 mg/L [28]. 

In 2014, a surveillance program was established by the national center of microbiology in Spain, and 273 samples of *Aspergillus* species were included, with 158 (58%) identified as part of the *A. fumigatus* complex [29]. MIC_50_ and MIC_90_ values for isavuconazole were, respectively, 1 and 4 for *A. fumigatus* complex and 1 and 2 for *A. flavus* and *A. terreus* complex. Susceptibility was defined according to EUCAST MIC breakpoints from 2018, i.e., for isavuconazole, resistance was considered from MIC > 1 for *A. fumigatus* and *A. terreus*, insufficient evidence for *A. flavus*. Mutations of environmental origin were detected in 4 *A. fumigatus* isolates (2.5%). Among 101 *A. fumigatus* complex isolates tested for isavuconazole susceptibility, 27 tested resistant to isavuconazole, with 17 isolates having no mutations in *cyp51A*, and 10 of them resistant only to isavuconazole. However, all these species had a MIC of 2 mg/L, which according to the 2020 EUCAST document corresponds to Areas of Technical Uncertainty (ATU) and is not considered resistant [30]. In fact, the following comment is provided in the EUCAST document: “if voriconazole wild-type (*A. flavus*: voriconazole MIC ≤ 2 mg/L; *A. fumigatus*: voriconazole MIC ≤ 1 mg/L) report as isavuconazole S and add the following comment: The MIC of 2 mg/L is one dilution above the S breakpoint but within the wild-type isavuconazole MIC range due to a stringent breakpoint susceptibility breakpoint” [30].

In conclusion, isavuconazole has very good activity against *Aspergillus* species, except for *A. fumigatus* isolates with resistance mutations of environmental origin. Higher doses might be effective in treatment of *Aspergillus* isolates with MIC of 2 mg/L. Good activity against most *Mucorales* has been detected, similar to that of posaconazole and slightly lower than L-AmB.

## 3. Pharmacokinetics and Pharmacodynamic of Isavuconazole

Isavuconazole is the active moiety of isavuconazonium sulfate, its water-soluble prodrug, which is hydrolyzed by plasma esterases (predominantly bybutyrylcholinesterase) following oral or intravenous administration. Each 372 mg of isavuconazonium sulfate corresponds to 200 mg of isavuconazole (2 capsules or 1 reconstituted vial). Recommended dosages are as follows: a loading dose of 200 mf isavuconazole every 8 h for 6 doses (48 h) followed by maintenance dose with 200 mg once daily starting 12 or 24 h following the last loading dose [21]. 

The bioavailability of isavuconazole is 98%, making the intravenous and oral formulations interchangeable and absorption is not altered by food intake [31]. Following oral administration, maximum plasma concentrations (C_max_) of isavuconazole are reached in 2–3 h and are dose-proportional. In a study in healthy volunteers, C_max_ after single oral administration of 200 mg of isavuconazole was 2.59 μg/mL (±0.449), similar to that measured after single intravenous administration—2.47 μg/mL (±0.374) [32]. Due to its long mean terminal half-life (100–130 h), isavuconazole accumulates in plasma with multiple dosing, as proved by a multiple-dose pharmacokinetic study in healthy volunteers, which estimated drug accumulation in plasma by dividing the AUC_0–24h_ on the last day of the study by the AUC_0–24h_ on day 1, both for the oral and intravenous formulations [33]. Similar results were obtained in a dose-escalation study of patients with acute myeloid leukemia and neutropenia [34]. 

Isavuconazole is largely (>99%) protein bound (compared to 58% of voriconazole), mainly to albumin, and has a large volume of distribution, as expected for a highly lipophilic agent [35]. Tissue penetration has proven to be widespread and rapid in animal models, reaching steady state within 14 days in a daily-dose regimen, comprising bone and brain tissues [36], while data regarding tissue penetration in humans are limited to case reports [37,38,39]. Low levels of isavuconazole in cerebrospinal fluid have been reported [36]. 

Analyses regarding population pharmacokinetics of isavuconazole during treatment of invasive fungal disease were retrieved from the two main trials assessing isavuconazole efficacy [40]. Desai et al. analyzed 6363 drug concentration values from 421 individuals comprising healthy volunteers and patients with invasive fungal infections included in the SECURE trial. Of note, all drug concentration values in the patient population were above the quantification limit and only 5% of those of the healthy volunteers’ population were below the quantification limit. When comparing the two populations, no difference in drug exposure was observed. Intersubject variability was calculated taking into account the area under the curve (AUC) value of isavuconazole and was 55% in the patient population and 36% in the healthy subject population [40]. Similar findings confirming low intersubject variability were reported by Kaindl et al. who analyzed isavuconazole plasma concentrations of samples drawn from patients included in a phase 3 SECURE trial [41], and by Kovanda et al. who estimated 63% of intersubject variability of isavuconazole clearance in 136 patients included in the VITAL study [42]. Another interesting finding from the analysis performed by Desai et al. was the effect of race on isavuconazole clearance, with Asians having 35% lower clearance compared to Caucasians [40]. This same finding was not confirmed in the study of Kovanda et al. with samples from the VITAL study, although the low prevalence of Asians and the overall sample size might have affected the results [42].

BMI was assessed as a possible factor influencing isavuconazole pharmacokinetics by Desai et al. and was found to be related to the volume of distribution of isavuconazole in the peripheral compartment, with obese (defined as BMI > 30) individuals having a greater volume of distribution when compared to non-obese ones [42], while in the study by Kovanda et al. BMI was reportedly associated with isavuconazole clearance. Weight was assessed as an independent covariate in the latter study and found to be associated both with clearance and with central volume distribution [40].

Estimated glomerular filtration rate (eGFR) was also assessed as a covariate in the study by Kovanda et al. and was found not to be associated with clearance of isavuconazole. All studies on TDM of isavuconazole are discussed in the dedicated paragraph. As other triazoles, isavuconazole is a substrate of cytochrome P450 (CYP450), particularly of isoenzymes 3A4 and 3A5. This characteristic is responsible for longer half-lives and reduced clearance in individuals with mild/moderate liver disease (Child–Pugh classes A and B), although no dose adjustment is yet suggested in this population [41]. No studies have assessed the use of isavuconazole in Child–Pugh class C patients.

Isavuconazole is a moderate inhibitor of CYP3A4 [43]. As such, drug–drug interactions with other CYP3A4 inducers/inhibitors are expected. Although weak, isavuconazole inhibitory activity on P-glycoprotein may cause clinically relevant interactions with digoxin; indeed, digoxin plasma levels monitoring is recommended in the case of co-administration with isavuconazole. The same mechanism accounts for potential drug–drug interactions with atorvastatin and metformin, whose plasma levels are increased when co-administered with isavuconazole, although no adverse effects were reported when studying co-administration in healthy subjects [44].

## 4. Isavuconazole in CAPA

For the purpose of this review, studies which reported the antifungals used for treatment of CAPA or CAM were identified and screened. We selected and focused on studies in which isavuconazole use was reported, either as monotherapy or in combination, in order to report the use pattern and rate. 

We identified 14 studies reporting the use of isavuconazole for CAPA, as shown in Table 2 [3,5,45,46,47,48,49,50,51,52,53,54,55,56]. In these 14 studies, 235 of 275 CAPA patients (85%) received antifungal treatment, whereas 40 patients did not receive any type of antifungal treatment. Early death was the main reason for not delivering antifungal medication. Out of 235 treated patients, 215 (91.5%) received first-line antifungal monotherapy: 107 (45.5%) with voriconazole, 64 (27.2%) with isavuconazole, and 44 (18.7%) with L-AmB. Overall, voriconazole was the most frequently used agent (155 patients, 66%), followed by isavuconazole. Isavuconazole was used in 95 (40%) of included patients: in 64 as first line monotherapy, while 22 received isavuconazole in combination and 9 received as sequential monotherapy, usually following voriconazole.

The largest multicenter cohort study reported 109 CAPA cases, and 99 patients received antifungal treatment [48]. Voriconazole was the most frequent monotherapy given to 53% of patients, followed by isavuconazole monotherapy given to 36% of subjects; treatment changes were not detailed but overall, 143 different treatment options were provided to 99 patients. 

In another multicenter study from France, 58 patients (76%) received various antifungal treatment regimens, with isavuconazole (not specified if in monotherapy or in combination) being given to 11 (19%) [49].

In a report from Brazil, 12 of 14 ICU CAPA cases received antifungal treatment, mainly voriconazole monotherapy (67%), while in 1 case L-AmB was changed to isavuconazole due to chronic renal failure [55]. On the contrary, in two small studies from Greece and Spain, isavuconazole monotherapy was the most frequently used (approximately 80% of patients) [50,51]. In the Greek experience, CAPA incidence rate in ICU was 3.3%, and mortality was as high as 67%, but only 1 patient died due to CAPA, while 3 died due to septic shock or multiorgan failure [50]. In the Spanish study, antifungal treatment was not provided to 3/8 patients diagnosed with CAPA due to their poor clinical conditions, while difficulties in performing TDM of voriconazole during the pandemic period made isavuconazole the preferred choice considering its attractive pharmacokinetic profile [51]. 

In one of the first studies reporting five CAPA cases in ICU in Germany, the only patient treated with isavuconazole was 70 years old and had acute renal failure necessitating slow low efficient daily dialysis (SLEDD) and elevated liver enzymes [8]. Similarly, in the study with 7 CAPA cases in ICU in Belgium, among 5 patients started on voriconazole, 2 were subsequently switched to isavuconazole, as both of them were receiving renal replacement therapy (RRT) and did not obtain detectable voriconazole serum levels [3].

To gain insight into the overall use of isavuconazole treatment in CAPA, including also the studies in which isavuconazole was not mentioned, we analyzed the data from two recent extensive reviews on CAPA. In the first one, Dimopoulos and colleagues reviewed 35 papers on CAPA treatment [56]. Out of 35 papers, 30 papers reported data on antifungal treatment and 7 papers reported data on isavuconazole treatment either as monotherapy or in combination. Among a total of 189 patients, 135 (71%), received antifungal treatment, mainly with voriconazole (35%), while 18 (13%) received isavuconazole: 9 as monotherapy, and 9 in combination with other antifungals. In the second review, Feys and colleagues identified and reviewed 48 papers on CAPA. Among them, 34 reported data on antifungal treatment, and a total of 429 of 820 (52%), patients were reported to have received antifungal treatment. In 12 papers, isavuconazole was used as CAPA treatment. Overall, 37 of 429 (8.6%) treated patients received isavuconazole: 23 (62%) as monotherapy, 5 in combination, and 9 as sequential monotherapy following voriconazole [5]. 

## 5. Isavuconazole in CAM

We identified 6 studies that reported the use of isavuconazole in CAM patients, either alone or in combination with other antifungals (Table 3) [13,14,58,59,60,61]. A total of 311 CAM cases were reported in these 6 studies and 304 of 311 (97.7%) received antifungal treatment. Among 304 treated CAM patients, 39 (13%) received isavuconazole: 28 as a monotherapy (with possible changes from or to other monotherapy regimens), and 11 in combination with other antifungals. Overall, 187 of 304 (62%) patients received combination antifungal therapy and 117 (38%) monotherapy. 

A single study reported 10 cases of rhino-orbital CAM in India [58]. All patients received antifungal treatment. Isavuconazole was used as monotherapy in one patient and in two others in combination with liposomal amphotericin B or amphotericin B deoxycholate. All 10 patients had uncontrolled diabetes mellitus. Out of the total patients, 9 survived and improved clinically and 1 patient expired due to secondary infection and sepsis. 

Thirteen patients with CAM were reported in six different centers in Germany. All the patients have underlying medical conditions except one. Among them, 12 received antifungal treatment. Isavuconazole was used in 6 patients; of them, 3 received isavuconazole as monotherapy and they had infection due to *R. microsporus*, *Lichtheimia*, and *Lichtheimia corymbifera* species. Of note, the most common Mucorales species reported in this study was *Rhizopus* spp. (10/13) with *Lichtheimia* and *Rhizomucor* being found in the other cases [61]. 

In the case of 4 CAM cases reported in the Netherlands, 3 in ICU and 1 outside ICU, all patients received combination antifungal therapy, and isavuconazole was used in 2 of them [59].

A multicenter retrospective study across India was conducted to evaluate epidemiology and outcomes among cases of CAM. CAM was diagnosed in 187 of 287 (65%) patients with mucormycosis. In comparison to patients with mucormycosis not associated with COVID-19, the administration of liposomal amphotericin B was lower in the CAM group (73% vs. 84%), while isavuconazole was used more frequently used in the CAM group (10% vs. 2%). Antifungal combinations, such as amphotericin B plus triazoles, were used much more frequently in CAM patients (50% vs. 12%) [14].

Finally, Hoenigl et al. reported 80 CAM cases from 18 countries, mainly from India. Liposomal Amphotericin B was used in the vast majority (89%) of patients, and posaconazole was used in addition to liposomal amphotericin B in 6 patients having rhino-orbital mucormycosis. Isavuconazole was given to 6 patients, as monotherapy in 1, in combination with L-amphotericin B in 3, and as salvage treatment in 2 [13]. 

In conclusion, among CAM cases reported from various parts of the world, the majority came from India, and combination treatment was used in most of the cases. 

## 6. Prophylaxis of CAPA in ICU

More research is needed to assess the need and efficacy of prophylaxis of CAPA in critically ill patients without an underlying malignancy or transplant. While initial experiences with CAPA prophylaxis have been published, a case–control retrospective study on posaconazole is ongoing (NCT05065658). For instance, in an observational study on the efficacy of prophylaxis for CAPA, 132 ICU patients with COVID-19, of whom 75 (57%) received antifungal prophylaxis (98% posaconazole, isavuconazole used in 1 patient due to impaired tolerance of posaconazole), were included. Of 10 CAPA cases diagnosed after a median of 6 days following ICU admission, 9 were recorded in the non-prophylaxis group. However, no difference in 30-day mortality was noted between the prophylaxis and no prophylaxis group (37% in both groups) [57].

In patients with high risk of invasive fungal disease, such as acute myeloid leukemia or transplant, antifungal prophylaxis has become a standard of care. Although isavuconazole does not have a formal indication for mold-active prophylaxis, it has been used in several studies, mainly in patients with hematological malignancies, and in lung transplant recipients in one study [33,62,63,64,65,66,67,68,69]. In these settings, the breakthrough rate of proven or probable IFD was in median 6%, ranging from 0 to 18% [33,62,63,64,65,66,67]. The breakthrough infections were also observed in patients with adequate isavuconazole levels, both in the aforementioned study and in a separate report [68,69].

Unfavorable patients’ characteristics, such as relapsed or refractory acute leukemia, might have contributed to rather high rate of breakthrough fungal infections in some of the initial studies (6.1–18%), since in the studies that included consecutive patients, the rate of breakthrough fungal infections was lower (0–3%) [33,65,66,69]. 

A randomized study is needed to prove the usefulness and efficacy of mold-active prophylaxis for CAPA with any agent, including isavuconazole.

## 7. Challenges of the Use of Isavuconazole in CAPA

The main challenges of the treatment of CAPA with isavuconazole stem from the uncertainty of the diagnosis and frequent lack of positive fungal culture, making the exclusion of azole-resistant aspergillosis less likely. However, with the application of recent diagnostic criteria, identification of patients with CAPA might be easier, with still some possibility of overdiagnosis, which can be reduced with repeated diagnostic testing [8]. Additionally, the use of molecular methods to detect resistance mutations of environmental origin is possible with some assays, and in that case, even an increased dose of azoles is not expected to be effective [70]. On the contrary, with wild type isolates, even with the MIC of 2 mg/L, high doses of isavuconazole might still be effective [28]. However, obtaining adequate plasma levels is required, since ICU patients receive frequently not only other drugs that may potentially have drug–drug interactions with azoles, but also undergo renal replacement therapy or extracorporeal membrane oxygenation (ECMO) which have been associated with a higher probability of subtherapeutic levels of azoles [71]. Therefore, even though TDM of isavuconazole is generally not recommended, the CAPA-specific guidelines recommended TDM of isavuconazole in selected cases [72]. 

## 8. Therapeutic Drug Monitoring (TDM) of Isavuconazole

Currently, there is still no consensus on the routine need for TDM of isavuconazole, based on data from the pivotal trial that did not find any association between blood levels and efficacy or toxicity of isavuconazole [73].

This might be due to the fact that approximately 10% of all 2458 blood level determinations from samples from clinical trials were >1 mg/L, and approximately 10% were >6 mg/L [74]. The distribution of blood levels and the rate of determinations < 1 mg/L were also similar in 283 real-life samples [74]. 

Although these rates of suboptimal levels are much lower than for other triazoles, no data was provided on predictors of suboptimal or very high blood levels of isavuconazole; therefore, several clinical studies explored this issue. We identified a total of 8 articles on isavuconazole TDM in various patient populations, which included overall data from 368 patients. The majority of included patients had hematological malignancy or were solid organ or hematopoietic stem cell transplant recipients, and a total of 1689 samples were taken for measurements of TDM (Table 4) [75,76,77,78,79,80,81,82]. 

In one of the first studies which included 264 isavuconazole blood concentrations from 19 patients, possible predictors of isavuconazole levels, such as weight, length of treatment, route of administration, and results of selected liver function tests, were analyzed in univariate and multivariate models. During the first 30 days of treatment, the median isavuconazole concentration in all patients was 3.69 mg/L (range 0.64–8.13 mg/L). Each day of treatment resulted in a statistically significant linear increase of 0.032 mg/L (range 0.023–0.041 mg/L). The link between the length of treatment and higher isavuconazole levels, and higher serum GGT and lower isavuconazole levels was confirmed in multivariate analysis. Six patients experienced adverse effects, the majority of which were gastrointestinal (32%). Isavuconazole was shown to be effective and well-tolerated, but prolonged use and high serum levels were associated with side effects, mostly gastrointestinal. The toxicity thresholds were 4.87 and 5.13 mg/L, respectively, based on time-dependent and fixed-time ROC curve analyses [76]. 

Long-term usage of isavuconazole in immunocompetent hosts is less well understood. In patients with chronic pulmonary aspergillosis, a retrospective study of isavuconazole TDM was performed, and mean levels of 4.1 mg/L were recorded in 45 patients. An even lower daily dose of isavuconazole (100 mg vs. 200 mg) resulted in satisfactory drug levels in a substantial number of patients, and it was also better tolerated and allowed therapy to be continued for longer periods of time. In that study, 16 patients (36%) discounted isavuconazole due to side effects [79].

In another experience in 32 adults with both malignant and non-malignant diseases, median blood levels were 2.35 mg/L. Low blood levels were noted in the case of treatment with Cytosorb, extracorporeal oxygenation (ECMO), and possibly also continuous renal replacement therapy (CRRT), particularly if given together, as was the case of 3 patients [81]. Among them, isavuconazole level was >1 µg/mL on day 12 of isavuconazole treatment in one and 1.7 mg/L in the second one; and in the third patient blood levels were <1 µg/mL on days 1 and 4 but increased to 2.44 mg/L when ECMO was stopped while RRT was continued. 

Risum et al. reported their data on 273 isavuconazole and 1242 voriconazole measurements. The median intra-/interindividual coefficient of variation was 43%/55% for isavuconazole compared to 53%/83% for voriconazole. The majority of patients had isavuconazole levels within what was considered a therapeutic range. The blood levels in patients with adverse events attributable to isavuconazole were similar to those without side effects [80].

The administration of isavuconazole capsules via an enteral feeding (EFT) tube in 19 transplant patients (15 lung transplants) was studied. TDM was performed after a median of 1 week (range 6–17 days) following EFT administration and 2 weeks (range 7–174 days) of isavuconazole therapy and achieved >1 mg/L in 17/19 patients (median value 1.8 mg/L, range 0.3–5.2). The levels were similar in those with and without previous intravenous administration [82]. 

Finally, Cojutti et al. reported the population pharmacokinetic and pharmacodynamic analysis of isavuconazole in a retrospective cohort of 50 hospitalized patients [78]. The risk of low through levels (<1.0 mg/L) was approximately 1%, and the risk of high levels (>5.13 mg/L) was 28% at 4 weeks and 39% at 60 days of therapy. Monte Carlo simulations showed that the standard daily maintenance dose of 200mg was adequate for achieving optimal cumulative fraction of response (defined as an AUC_24h_/MIC > 33.4, which is the pharmacodynamic index of efficacy based on EUCAST clinical breakpoint) against *A. fumigatus* and *A. flavus* with MIC up to 1 mg/L. Cumulative fractions of response were >90% in the first two months of treatment.

In addition, the use of high-performance liquid chromatography (HPLC) assay for routine determination of isavuconazole blood levels was studied and validated [83].

In conclusion, most of the patients in real-life settings managed to obtain adequate blood levels of isavuconazole. Therefore, there is lower necessity of TDM for isavuconazole, compared to voriconazole and posaconazole. Blood levels increase with prolonged use, so dose reduction might be required in the case of prolonged (several months) treatment. In the case of CAPA or CAM, this is an unlikely scenario, but certain ICU-specific treatments, such as ECMO or RRT might be associated with inadequate levels warranting TDM in this setting.

## 9. The Impact of Obesity on Isavuconazole Levels

Since obesity is a well-recognized risk factor for severe COVID-19, including ARDS and ICU admission, obese patients were frequent among subjects with CAPA. Therefore, treatment options against CAPA should consider the impact of body mass index (BMI) on their efficacy. Although different studies were performed on the impact of obesity on isavuconazole levels, no clear answers were provided. In the study by Desai et al. performed in liver disease patients and healthy controls, BMI had a direct influence on isavuconazole peripheral volume of distribution, but BMI did not affect the exposure of isavuconazole [40]. In another study by Kovanda et al., both weight and BMI enhanced the clearance of isavuconazole; however, only weight had a direct relation with the volume distribution of the central compartment, and these results were partially contradictory to the study performed by Desai et al. which showed no relationship of weight with isavuconazole pharmacokinetics [42]. Another study was performed in solid organ transplant patients treated with intravenous isavuconazole and reported that patients with BMI ≥ 18.5 kg/m^2^ showed 48% lower area under the concentration time curve (AUC) than the patients with BMI < 18.5 kg/m^2^ [84]. These findings were comparable to the AUC in myeloid leukemia patients [33]. Patients with lower BMI (BMI < 18) had a higher risk of supratherapeutic isavuconazole concentrations as compared to those with BMI ≥ 18.5 kg/m^2^, who were at risk of subtherapeutic isavuconazole [84]. 

In conclusion, BMI might influence the exposure to isavuconazole suggesting that patients with very high BMI might have lower therapeutic levels, while underweight patients might have higher isavuconazole exposure.

## 10. The Impact of Renal Replacement Therapy, Hemodialysis, and ECMO on Isavuconazole Therapy 

Critically ill patients frequently need supportive therapy for renal or respiratory failure, such as CRRT and ECMO. 

Similarly to other antifungals, isavuconazole exposure is not affected by renal impairment and no dose adjustment is needed, even in the case of hemodialysis. Unlike for amphotericin B, renal toxicity is not a side effect of isavuconazole. Unlike for voriconazole and posaconazole, the intravenous formulation of isavuconazole does not contain cyclodextrin; therefore, there is no risk of accumulation of this intravenous vehicle (SBECD) in patients with renal replacement, and both oral and intravenous formulations may be used in patients with renal impairment. Isavuconazole is not readily dialyzable. A dose adjustment is not warranted in patients with end-stage renal disease. As far as CRRT is concerned, for both continuous veno-venous hemofiltration and hemodiafiltration, a standard dose should be used. Of note, echinocandins such as caspofungin or micafungin require a higher loading dose in the case of CRRT. However, lower plasma levels of isavuconazole have been reported in a patient with CRRT (see also Table 4) [81]. Indeed, in 7 patients receiving RRT, the median level of isavuconazole was 0.91 mg/L, ranging from 0.66 to 2.44, with IQR of 0.82–1.36 mL/L. Even after excluding measurement from patients treated also with ECMO or Cytosorb^®^ adsorber, the median isavuconazole levels were still below 1: 0.91 mg/L, ranging from 0.75 to 2.44, with IQR of 0.90–1.36 in patients with RRT.

Additionally, patients treated with ECMO had suboptimal isavuconazole levels, never reaching 1 mg/L in 1 patient, but increasing to 2.44 after discontinuing ECMO and still receiving RRT [81]. In another case of a patient with pulmonary blastomycosis, for which therapeutic target concentration of isavuconazole was set at 3 mg/L, these concentrations could be only achieved by increasing the dose of isavuconazole (200 mg bid), while the standard dose resulted in a concentration of 1.9 mg/L [85]. In an additional case of a patient treated with ECMO but no RRT, treatment with voriconazole did not result in adequate plasma levels, while standard dose of isavuconazole resulted in initial adequate plasma level (1.7 mg/L), which nevertheless decreased to 0.7 mg/L over the following 12 days [86]. The ECMO circuit exchange (which was done approximately every 7 days) was cited as the most probable cause of the decrease in plasma levels. When double dose of isavuconazole was administered daily, isavuconazole concentration remained stable during the rest of treatment with levels of 3.7 and 2.9 mg/L. Of note, it is well known that dose of voriconazole and L-AmB should be increased in the case of ECMO, while contrasting data have been reported for posaconazole [87].

In conclusion, CRRT and ECMO may result in subtherapeutic levels of isavuconazole; thus, TDM is required in such cases. 

## 11. Drug–Drug Interactions (DDIs)

Due to increasing usage of isavuconazole in COVID-19 patients in the context of CAPA, drug–drug interactions between isavuconazole and COVID-19 treatments are a subject of interest. Of note, this topic is still largely unexplored. 

Remdesivir, the prodrug of the adenosine analogue GS-441524, is a substrate and a weak inhibitor of CYP3A4 and a substrate of P-glycoprotein [88], both moderately and weakly inhibited by remdesivir, respectively. Although no large study has evaluated potential DDIs between these two drugs, these are unlikely to be expected, as remdesivir metabolism has been predicted to be insignificantly affected in in vivo prediction models even by complete inhibition of CYP3A [89]. The same is true for molnupiravir, another nucleoside analogue, for which no DDIs are expected given that it neither inhibits nor induces any CYP enzyme [90]. 

On the contrary, DDIs are expected with the boosted protease inhibitor Paxlovid^®^ (nirmatrelvir/ritonavir), due to the ritonavir component, a well-known strong CYP3A4 inducer. DDIs of ritonavir with isavuconazole had already been assessed in the combination with lopinavir with the evidence of a significant increase in the C_max_ and AUC of isavuconazole and decrease in the C_max_ and AUC of lopinavir and ritonavir [91]. These data had led to contraindication of coadministration of high-dose ritonavir (>200 mg/day) and isavuconazole reported in the isavuconazole prescription information, while no dose adjustment is indicated for the 200 mg/day ritonavir dosage, although monitoring for DDIs and antiviral inefficiency is warranted [92]. The same degree of DDIs could be expected with ritonavir-boosted nirmatrelvir, itself metabolized by CYP3A4 in in vitro studies, although no data on coadministration is available. Currently, the FDA-fact sheet on Paxlovid^®^ contraindicates coadministration with voriconazole, while it refers to isavuconazole product labels for coadministration with isavuconazole [93]. Given the indicated dosage of nirmatrelvir/ritonavir (300 + 100 mg or 150 + 100 mg BID based on renal function) and the short duration of coadministration (5 days), no significant adverse effects should be expected, and after stopping antiviral, the CYP3A4 inhibitory effect is likely to disappear within 3 days. The role of close monitoring of possible adverse effects related to increased plasma concentrations of isavuconazole and TDM of isavuconazole to prevent toxicity is unclear due to short term co-administration and lack of clear cut-off for severe acute toxicity of isavuconazole. However, the antiviral activity can be expected to be lower due to decreased plasma levels of nirmatrelvir/ritonavir, which can be of particular significance in the case of immunocompromised patients, who are frequently receiving isavuconazole treatment and are commonly subject to nirmatrelvir/ritonavir prescription, possibly reducing the efficacy of nirmatrelvir/ritonavir in the prevention of severe COVID-19 in this high-risk population. 

## 12. Conclusions

Isavuconazole therapy has been used only in a minority (less than 10%) of all published cases of patients with CAPA. However, when analyzing only reports that included patients treated with isavuconazole, thus assuming with certainty the availability and expertise in the use of this therapeutic option, isavuconazole was the second most frequently prescribed monotherapy agent after voriconazole, reported to be used as first-line monotherapy in 27% of patients. CAM patients were mainly treated with isavuconazole in combination with amphotericin B, and only rarely (9%) treated with isavuconazole monotherapy. The challenges of the use of isavuconazole in CAPA and CAM depend on obtaining adequate exposure in COVID-19 patients who are frequently obese and admitted to ICU, where they may receive RRT or ECMO. For these reasons, and given the difficulties in assessing the response to antifungal treatment in the case of co-existing pulmonary infection, TDM of isavuconazole in these cases might be warranted in order to confidently proceed with isavuconazole antifungal treatment. On the other hand, the advantages of the use of isavuconazole in this setting include the availability of data on high efficacy in the treatment of aspergillosis in the immunocompromised, lower potential for DDIs and no risk of QT prolongation compared to other mold-active azoles, better security profile than voriconazole, and the possibility of the use of intravenous formulation in the case of renal failure. Dedicated studies on the efficacy of isavuconazole monotherapy in CAM and CAPA are warranted. 

## Figures and Tables

**Table 1 jof-08-00674-t001:** MIC_50_, MIC_90_, ECOFF, and breakpoint values (mg/L) for isavuconazole and comparators.

Study, Species (Number of Isolates Tested)	Isavuconazole		Voriconazole		Posaconazole	
	MIC_50_, MIC_90_	ECOFF/R Breakpoint According to EUCAST v. 10.0 2020	MIC_50_, MIC_90_	ECOFF/R Breakpoint According to EUCAST v. 10.0 2020	MIC_50_, MIC_90_	ECOFF/R Breakpoint According to EUCAST v. 10.0 2020
Pfaller et al., 2021 [23]						
*A. fumigatus* ISA (n = 70), VORI (n = 189), POSA (n = 189)	0.5, 1	2/>2	0.5, 0.5	1/>1	0.25, 0.5	0.25/>0.25
*A. flavus* ISA (n = 19), VORI (n = 43), POSA (n = 43)	0.5, 1	2/>2	0.5, 1	2/-	0.25, 0.5	0.5/-
*A. niger* ISA (n = 18), VORI (n = 46), POSA (n = 46)	0.5, 4	4/-	1, 1	2/-	0.5, 1	0.5/-
*A. terreus* ISA (n = 6), VORI (n = 14), POSA (n = 14)	2, ND	1/>2	0.25, 0.25	2/-	0.25, 0.25	0.25/>0.25
*A. nidulans* ISA (n = 6), VORI (n = 11), POSA (n = 11)	0.03, ND	0.25/>0.25	0.12, 0.25	1/>1	0.25, 0.5	0.5/-
Astvad et al., 2017 [22]						
*A. fumigatus* (n = 211)	1, ND	2/>2	0.5, ND	1/>1	ND	0.25 > 0.25
*A. niger* (n = 41)	2, ND	4/-	1, ND	2/-	ND	0.5/ND
*A. terreus* species complex (n = 27)	1, ND	1/>2	1, ND	2/-	ND	0.25/>0.25
*A. flavus* species complex (n = 19)	1, ND	2/>2	1, ND	2/-	ND	0.5/-

Abbreviations: ECOFFs, epidemiologic cut-off value; EUCAST, European Committee on Antimicrobial Susceptibility Testing; MIC_50_, Minimum Inhibitory Concentration required to inhibit the growth of 50% of organisms; MIC_90_, Minimum Inhibitory Concentration required to inhibit the growth of 90% of organisms; ISA, isavuconazole; ND, no data; POSA, posaconazole; VORI, voriconazole; -, not defined.

**Table 2 jof-08-00674-t002:** Fourteen studies which reported patients with CAPA in whom isavuconazole treatment was used.

References	Country	Total No of Patients with CAPA	No of the Patients Who Received Antifungal Treatment	Treated with ISA Monotherapy (Usually as 1st Line)	Treated with VORI Monotherapy (Usually as 1st Line)	Treated with L-AmB Monotherapy (Usually as 1st Line)	Other Antifungals in Monotherapy	Combined or Sequential Treatment	Outcome in ISA-Treated Patients
Falces et al., 2020 [46]	Spain	10	8	None	2/8 (25%)	None	2/8 (25%) AmB	VORI + CASP in 1 AmB followed by ISA in 1 Sequential combination treatment: MICA + VORI > AmB + ISA in 1 ANID followed by AmB in 1	No data
Gangneux et al., 2020 [47]	France	9	7	All treated with VORI or ISA, no further details	None	None	None		No data
Rutsaert et al., 2020 [3]	Belgium	7	6	2/6 (33%)	4/6 (67%)	None	None	None	2/6 patients died; no data on which treatment they received
Koehler et al., 2020 [8]	Germany	5	5	1/5 (20%)	2/5 (40%)	None	2/5 CASP followed by VORI	None	The only patient treated with ISA died, no cause of death provided
Antinori et al., 2020 [45]	Italy	1	1	1/1 (100%)	None	None	None	-	Died soon after starting treatment
Prattes et al., 2021 [48]	Europe, USA, Pakistan	109	99	36/99 (36%) *	52/99 (53%)	17/99 (17%)	POSA 4/99 (4%), Echinocandins 13/99 (13%), Deoxycholate AmB 3/99 (3%)	18/99 (18%) VORI or ISA combined with echinocandin or L-AmB	No data
Lahmer et al., 2021 [53]	Germany	11	11	1/11 (9%)	5/11 (45%)	5/11 (45%)	None	None	No data for ISA treated patient
Hatzl et al., 2021 [57]	Austria	9	9	3/9 (33%)	None	None	6/9 (67%) POSA	None	No data for 3 ISA treated patients
Fekkar et al., 2021 [52]	France	7	6	None	None	None	CASP	VORI + CASP in 3 L-AmB + CASP in 1 VORI/L-AmB/CASP/ISA in 1 VORI/CASP/L-AmB in 1	Success in the single patient who received combination treatment including ISA
Paramythiotou et al., 2021 [50]	Greece	6	6	5/6 (83%)	None	None	CASP	Sequential treatment CASP > L-AmB in 1	Two patients alive at the last follow up and still on ISA treatment, 3 patients died, mainly due to MDR *A. baumanni* infection
Machado et al., 2021 [51]	Spain	8	5	4/5 (80%)	None	1/5 (20%)	None	None	All 5 died due to CAPA
Wasylyshyn et al., 2021 [54]	UK	3	2	None	None	None	None	Sequential treatment VORI > ISA in both	Both ISA treated patients alive at 12 weeks.
Gangneux et al., 2022 [49]	France	76	58 (76%)	11/58 (19%) *°	44/58 (76%)	20/58 (28%)	CASP 16 (28%), unspecified 5 (9%)	29/58 more than one type	No data
de Almeida et al., 2022 [55]	Brazil	14	12	None	8/12 (%)	1/12 (%)	-	2 days of L-AmB + ISA, followed by ISA in 1 Sequential monotherapy VORI > L-AmB and L-AmB > VORI in 2	The only patient treated with a combination containing ISA: died on day 19 after diagnosis of CAPA; no direct cause of death provided

* changes or sequential therapy details not available; ° not specified if all given in monotherapy. Abbreviations: ANID, anidulafungin; AmB, amphotericin B; CASP, caspofungin; ISA, isavuconazole; L-AmB, liposomal amphotericin B; MICA, micafungin; POSA, posaconazole; UK, United Kingdom; USA, United States of America; VORI, voriconazole.

**Table 3 jof-08-00674-t003:** Studies reporting patients with CAM in whom isavuconazole treatment was used.

References	Country	Total No of Patients with CAM	No of the Patients Who Received Antifungal Treatment	Treated with ISA Monotherapy	Treated with (L-) AmB Monotherapy	Treated with POSA Monotherapy	Other Antifungals Monotherapy	Combined Treatment	Outcome in ISA-Treated Patients
Patel et al., 2021 [14]	India	187	187	19/187 (10%) *°	136/187 (73%) *°	73/187 (39%) *°	AmB 31/187 (17%) *°	Single antifungal 95/187; combination therapy 137/187; sequential 79/187	No such data was provided on patients’ responses to ISA. However, survival rate was high when patients were receiving antifungal drugs concurrent, sequential, and medical surgery
Hoenigl et al., 2021 [13]	18 Countries	80	79	3/79 1st line in 1; salvage therapy in 2	54/79 (68%) *°	6/79 (8%) *°	CASP, VORI and MICA	Antifungal combination 14/79 3 patients’ combination therapy ISA + L-AmB	
Arjun et al., 2021 [58]	India	10	10	1 (10%)	6 (30%) L-AmB or d-AmB	None	None	L-AmB + ISA in 1; d- AmB + ISA in 1; L-AmB/d-AmB followed by ISA	All the three ISA-treated patients were improved and discharged
Buil et al., 2021 [59]	The Netherlands	4	4	None	None	None	None	VORI (days 0–13); L-AmB (from day 13); POSA (from day 19) in 1; VORI (7–12); L-AmB (12–23) in 1; VORI + ANID (0–21); POSA (13–23); ISA (21–24); VORI (24–30); ISA (30–35); ISA + L-AmB + INF-γ (35–43); AmB bladder irrigation (39–43) in 1; L-AmB + ISA + INF-γ for 7 weeks in 1	Two patients received combined treatment of ISA and L-AmB, and died due to CAM
Danion et al., 2022 [60]	France	17	12	2/12 (17%)	10/12 (83%)	None	None	No combination treatment	One patient was alive after receiving 3 months treatment of ISA and one died, no cause of death provided
Seidel et al., 2022 [61]	Germany	13	12	3/12 (25%)	2/12 (17%)	None	Echinocandin 1/12 (8%)	ISA + L-AmB + VORI in 3; ISA + L-AmB in 1; VORI + echinocandin in 1; ISA + echinocandin in 1	No data

* changes or sequential therapy details not available; ° not specified if all given in monotherapy. Abbreviations: AmB, amphotericin B; ANID, anidulafungin; CAM, COVID-19-associated mucormycosis; CASP, caspofungin; d-AmB, deoxycholate amphotericin B; INF-γ, interferon gamma; ISA, isavuconazole; L-AmB, liposomal amphotericin B; MICA, micafungin; POSA, posaconazole; VORI, voriconazole.

**Table 4 jof-08-00674-t004:** Main studies reporting the experience with TDM of isavuconazole.

References	Patients’ Underlying Condition (Number)	Total No. of Patients	No. of Measurements	Mean/Median, mg/L	Min–Max, mg/L	Subtherapeutic Levels (<1 mg/L)	Potentially Supratherapeutic Levels	Safety: No. of Patients with Side Effects	Comment
Furfaro et al., 2019 [76]	HM (13); other (6)	19	264	Median 3.6; median 2.86 during the first 14 days; median 4.4 after 14 days of therapy	0.64–8.13	ND	ND	6 (31.6%) gastrointestinal	Failure in 1 with concentration of 1.55 GI side effects were associated with high levels (5.13 mg/L) and prolonged administration Drug accumulation observed over time
Kosmidis et al., 2020 [79]	Chronic pulmonary aspergillosis	45	285	Overall mean 4.1; mean 4.6 if dose 200 mg/day; mean 4.1 if 200 mg/100 mg on alternate days; mean 3.7 if 100 mg/day	1.1–10.1	<1 mg/L in none of the patients All 117 measurements from patients taking 100 mg/day were >1 mg/L	>6 mg/L in 36 (13%)	16 (36%) discounted ISA due to side effects (5 within 28 days) such as hepatotoxicity in 4, neuropathy in 3, headache in 2, malaise in 2, weight loss in 1, confusion in 1, nausea in 1, photosensitivity in 1 case, dysgeusia in 1	38 patients (86%) were started on a standard dose ISA at lower dose resulted in acceptable levels and favorable profile during > 6 months of therapy
Borman et al., 2020 [75]	ND	150	210	Mean 3.32	0.5–11.6	<1 mg/L in 6 (4%), patients 62 (41%) achieved target level of 2–4 mg/L	ND	ND	In patients <18 years greater interpatient variability of blood levels was found
Zurl et al., 2020 [81]	HM (14), SOT (4), cancer (2), other (12), including osteomyelitis	33	140	Median 2.35 If RRT/ECMO excluded: median 3.05	0.66–9.1 1.38–9.1	Only in case of RRT, ECMO, or Cytosorb use	ND	6 (18%) developed side effects: 1 anaphylaxis, 1 leukopenia, 2 increased liver enzymes, 1 paraesthesia, 1 erythema, and elevated liver enzymes	Lower concentration in case of RRT (median 0.91 in 7 patients), ECMO, and Cytosorb^®^
McCreary et al., 2020 [82]	SOT (18), HSCT (1) treated with ISA via enteral feeding tube	19	ND	Mean 1.8	0.3–5.2	<1 mg/L in 2	>5 mg/L in 1	ND	Favorable PK confirms that capsule content can be safely sprinkled into an enteral feeding tube
Risum et al., 2021 [80]	HM (16); SOT (2); pulmonary disorder (13); (COPD in 7); other (5)	36	273	Median 4.3	0.5–15.4	<0.2 mg/L in 7 (no data on compliance) 32/273 (12%) measurements <2 mg/L	>10 mg/L in 9/247 (4%)	ND	One case of ISA detectable for 35 days after stopping
Kronig et al., 2021 [77]	All HM or HSCT	16	35	Mean 2.9	0.9–6.7	ND	ND	Discontinued in 5 (16%): hypersensitivity in 2, increased liver enzymes in 2, drug interactions in 1	
Cojutti et al., 2021 [78]	Onco-hematological malignancy (25); other (25)	50	199	Median 3.68	2.07–5.38	ND	ND	ND	Drug accumulation observed over time In Monte Carlo simulations standard dose was optimal against *A. fumigatus* and *A. flavus* with MIC up to 1 mg/L

Abbreviations: ECMO, extracorporeal membrane oxygenation; GI, gastrointestinal; HM, hematological malignancies; HSCT, hematopoietic stem cell transplantation; ISA, isavuconazole; MIC, minimum inhibitory concentrations; ND, no data; PK, pharmacokinetics; RRT, renal replacement therapy; SOT, solid organ transplant.

## Data Availability

Not applicable.

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
