# Peer review of "Isavuconazole for COVID-19-Associated Invasive Mold Infections"

_jof, 2022, doi:10.3390/jof8070674_

Round 1
Reviewer 1 Report
The focus of this article is primarily on reviewing the use of the triazole isavuconazole in patients with either aspergillosis or mucormycosis and COVID-19. Therefore the title of the paper would be better if the word fungal was replaced by mould (mold).
The abstract is appropriate to the topic covered.
There follows an Introduction which covers relevant areas concerning aspergillosis, mucormycosis, their occurrence in COVID-19 patients, and the use of isavuconazole for these infections.
Note a typo which should read European Medicines Agency (line 84).
There follows a rather lengthy section on the antifungal activity of isavuconazole.
I suggest removing reference to the drug's activity against Candida species as this is not the topic being covered.
I also suggest that for the reader the listing of lots of MICs is difficult to digest in the form presented.
It would be better to tabulate the MIC/ECOFF MIC50 data instead.
The word 'was' should be removed from line106
Proper names of organisms should be italicised (see line 115)
The sentence beginning "Since 2010...' on line 108 should be rephrased.
Change increasing to increasingly on line 123.
If there is a difference between 'strains' and 'isolates' then the authors should define what that difference is.
In the paragraph on azole-resistant Aspergillus strains, the authors suggest that "...resistance is thought to have been induced in the environment through the use of compounds similar to medical azoles..." A reference should be provided for this statement and I suggest the word induced should be changed to selected because the resistant variants may have always been present and have emerged under the selective pressure of agricultural azoles.
The sentence on line 126 is not clear as to its meaning
Note typo line 145.
Line 148 insert the before national
line 154 MIC of 0.1? Is this mg/L?; same line change was to were
Line 168 change have to has.
line 172 is butylchilesterase correct?
Line 211 change with respect to compared
The authors discuss the use of isavuconazole in CAPA and a linked Table 1, followed by its use in CAM accompanied by Table 2. The information provided is informative but the absence of any data on therapeutic outcomes seems to me an omission. I accept that there are limited data but it should be possible, from each of the studies where isavuconazole was used, to provide results on how many patients responded. This could be an additional column for each of the tables.
Table 3 lists studies where isavuconazole has been used for prophylaxis. Most of these appear to have been in haematological malignancy patients. It is not clear what the relevance of this is to COVID-19? Nevertheless, if they explain why this has been included then it is reasonable to retain it in the article.
On line 367 at the bottom of the table 3 they refer to 'a high rate' without explaining what it is they are talking about
The later sections of the article cover useful areas including TDM, impact of renal replacement therapy, obesity and drug-drug interactions
In their concluding section there are several typos e.g., prolungation, voricoanzole.
They finish with a surprising statement that: "Further studies on the efficacy of isavuconazole monotherapy in CAM are warranted" This suggests that there are adequate data in CAPA which I think is not the case. Furthermore, they haven't really considered therapeutic efficacy of isavuconazole in this article (as mentioned above).
References should be checked for omissions. For example reference 13 is incomplete.
Author Response
Dear Reviewers,
Thank you for giving us the opportunity to submit a revised draft of our manuscript titled [Isavuconazole for COVID-19 associated invasive fungal infections] to [Journal of Fungi]. We appreciate your time and effort in giving us important feedback on our article. The reviewers are to be thanked for their helpful remarks on our article. We integrated the suggested improvements that reflected the majority of the reviewers' comments. The modifications in the document have been done with track changes.
Here is a point-by-point response to the reviewers’ comments and concerns.
Comment 1: [The focus of this article is primarily on reviewing the use of the triazole isavuconazole in patients with either aspergillosis or mucormycosis and COVID-19. Therefore the title of the paper would be better if the word fungal was replaced by mould (mold).
Response: Thank you for pointing this out. We agree with this comment. The title was modified to (Isavuconazole for COVID-19 associated invasive mould infections).
Comment 2:[Note a typo which should read European Medicines Agency.
Response: Corrected
Comment 3: [There follows a rather lengthy section on the antifungal activity of isavuconazole. I suggest removing the reference to the drug's activity against Candida species as this is not the topic being covered. I also suggest that for the reader the listing of lots of MICs is difficult to digest in the form presented. It would be better to tabulate the MIC/ECOFF MIC50 data instead
Response: We followed the reviewer’s suggestion and deleted the activity against Candida and left in a new table 1 the activity against moulds, and Mic 90 and ECOFF were added.
Comment 4: The word 'was' should be removed from line106
Response: Corrected
Comment 5: Proper names of organisms should be italicised (see line 115)
Response: Corrected and changed
Comment 6: The sentence beginning "Since 2010...' on line 108 should be rephrased.
Response: Rephrased.
Comment 7: Change increasing to increasingly on line 123.
Response: Corrected
Comment 8: If there is a difference between 'strains' and 'isolates' then the authors should define what that difference
Response: We used the terms interchangeable, but for clarity we replaced the term “strains” with “isolates”
Comment 9: In the paragraph on azole-resistant Aspergillus strains, the authors suggest that "...resistance is thought to have been induced in the environment through the use of compounds similar to medical azoles..." A reference should be provided for this statement and I suggest the word induced should be changed to selected because the resistant variants may have always been present and have emerged under the selective pressure of agricultural azoles.
Response: Reference is added for the given statement and the word induced was replaced by selected
Comment 10: The sentence on line 126 is not clear as to its meaning
Response: The sentence has been corrected
Comment 11: Note typo line 145.
Response: Typo errors are corrected
Comment 12: Line 148 insert the before national
Response: Corrected
Comment 13: line 154 MIC of 0.1? Is this mg/L?; same line change was to were
Response: Not 0.1 1mg/L was changed to were
Comment 14: Line 168 change have to has.
Response: Corrected
Comment 15: line 172 is butylchilesterase correct?
Response: Corrected butyrylcholinesterase
Comment 16: Line 211 change with respect to compared
Response: Corrected
Comment 17: The authors discuss the use of isavuconazole in CAPA and a linked Table 1, followed by its use in CAM accompanied by Table 2. The information provided is informative but the absence of any data on therapeutic outcomes seems to me an omission. I accept that there are limited data but it should be possible, from each of the studies where isavuconazole was used, to provide results on how many patients responded. This could be an additional column for each of the tables.
Response: Unfortunately, in studies that included several patients the data on outcome for single treatment were not provided, except for few ones. We added additional column (outcomes in isavuconazole treated patients in table 2 and 3 new numbers), so it is clear also to the readers that these are are at best partial.
Table 3 lists studies where isavuconazole has been used for prophylaxis. Most of these appear to have been in haematological malignancy patients. It is not clear what the relevance of this is to COVID-19? Nevertheless, if they explain why this has been included then it is reasonable to retain it in the article.
Response: We agree that all but one studies were performed in different populations. We removed the table from the paper
Comment 18: On line 367 at the bottom of the table 3 they refer to 'a high rate' without explaining what it is they are talking about
Response: We modified the sentence for clarity al follows: Unfavorable patients’ characteristics, such as relapsed or refractory acute leukaemia, might have contributed to rather high rate of breakthrough fungal infections in some of the initial studies (6.1-18%), since in the studies that included consecutive patients, the rate of breakthrough fungal infections was lower (0-3%) [33,65,66,69].
Comment 19: In their concluding section there are several typos e.g., prolungation, voricoanzole.
Response: Corrected
Comment 20: They finish with a surprising statement that: "Further studies on the efficacy of isavuconazole monotherapy in CAM are warranted" This suggests that there are adequate data in CAPA which I think is not the case. Furthermore, they haven't really considered therapeutic efficacy of isavuconazole in this article (as mentioned above).
Response: We changed the conclusion for clarity: Dedicated studies on the efficacy of isavuconazole monotherapy in CAM and CAPA are warranted.
Comment 21: References should be checked for omissions. For example reference 13 is incomplete
Response: Corrected
Reviewer 2 Report
This is an outstanding and much needed review on the role of isavuconazole for the treatmnet of COVID associated fungal infections. The topic has been investigated in great detail under supervision of Dr. Mikulska, a leading expert in the field, and the outcome is a prodcut that the authors should be proud of. The tables are outstanding. I definitely plan to use this review in many of my future presentations and papers as reference.
Only one very minor comment:
- Please replace reference 13 "Hoenigl, M.; Seidel, D.; Carvalho, A.; Rudramurthy, S.M.; Arastehfar, A.; Gangneux, J.P.; Nasir, N.; Bonifaz, A.; Araiza, J.; Klimko, N. The emergence of COVID-19 associated mucormycosis: analysis of cases from 18 countries. 2021"
with the published version in Lancet Microbe:
- "The emergence of COVID-19 associated mucormycosis: a review of cases from 18 countries." Lancet Microbe. 2022 Jan 25. doi: 10.1016/S2666-5247(21)00237-8. .PMID: 35098179
Author Response
Dear Reviewer,
Thank you for giving us the opportunity to submit a revised draft of our manuscript titled [Isavuconazole for COVID-19 associated invasive fungal infections] to [Journal of Fungi]. We appreciate your time and effort in giving us important feedback on our article. The reviewers are to be thanked for their helpful remarks on our article. We integrated the suggested improvements that reflected the majority of the reviewers' comments. The modifications in the document have been done with track changes.
Here is a point-by-point response to the reviewers’ comments and concerns.
Comment 1: Please replace reference 13 "Hoenigl, M.; Seidel, D.; Carvalho, A.; Rudramurthy, S.M.; Arastehfar, A.; Gangneux, J.P.; Nasir, N.; Bonifaz, A.; Araiza, J.; Klimko, N. The emergence of COVID-19 associated mucormycosis: analysis of cases from 18 countries. 2021 with the published version in Lancet Microbe
Response: Corrected and added the new published version of lancet microbe
Round 2
Reviewer 1 Report
The authors have satisfactorily addressed my comments and recommendations in order to improve the manuscript